# Measurement of Gas Flow Rate at Gasification of Low-Melting Materials in a Flow-Through Gas Generator

Dmitry A. Vnuchkov [1], Valery I. Zvegintsev [1], Denis G. Nalivaichenko [1] and Sergey M. Frolov [2,*]

1    Khristianovich Institute of Theoretical and Applied Mechanics of the Siberian Branch of the Russian Academy of Sciences (ITAM SB RAS), 630090 Novosibirsk, Russia
2    Department of Combustion and Explosion, Semenov Federal Research Center for Chemical Physics of the Russian Academy of Sciences, 119991 Moscow, Russia
*    Correspondence: smfrol@chph.ras.ru

**Abstract:** A semi-empirical method is proposed for determining the rate of gas production in a flow-through gas generator (GG) with the allocation of a part of the gas flow produced by gasification of a low-melting solid material (LSM) in the total gas flow rate through the GG. The method is verified by test fires with polypropylene sample gasification by hot air under conditions of incoming supersonic flow with Mach number 2.43, 2.94, and 3.81 and stagnation temperature 600–700 K. The mean flow rates of gasification products obtained in test fires were 0.08 kg/s at Mach 2.43, 0.10 kg/s at Mach 2.94, and 0.05–0.02 kg/s at Mach 3.81. For obtaining 1 kg of gasification products in the test fires there was a need of 1.61 to 2.86 kg of gasifying agent.

**Keywords:** flow-through gas generator; low-melting solid material; combustion; gasification

## 1. Introduction

Gas generators (GG) are designed for generating gases of desired composition, flow rate, temperature, and pressure through different physical and chemical processes such as liquid/solid charge vaporization, pyrolysis, gasification, combustion, etc. All GGs can be divided into two groups: closed-type GGs and flow-through GGs. In the former, a working medium is initially enclosed in a single housing, so such GGs are mostly single-action devices. In the latter, a working medium is continuously supplied to GG, which ensures long-term operation. The closed-type GGs include GGs deploying airbags in automobiles, injecting inhibitors in accidental fires, driving turbopumps in rocket engines, etc. [1–8].

The flow-through GGs are used in industry for production of various chemicals and energy, e.g., by converting crude oil, biomass, sewage sludge, municipal solid wastes, etc. to syngas, which is further used for production of hydrogen, synthetic natural gas, ammonia, methanol, Fisher–Tropsch diesel, and/or generating heat and electricity [9–12]. Given the widespread use of GGs, the methods for determining their characteristics at both design and operation stages is relevant.

For this purpose, methods of numerical simulation of physicochemical processes in GGs are now widely used [13–18]. However, at present the complexity of these processes does not allow predicting the main characteristic of GGs, i.e., the instantaneous rate of gas production, with sufficient accuracy. In view of this, experimental verification and refinement of numerical simulations remain important [19,20]. A good illustrative example is the GG of a hybrid rocket. Contrary to solid rockets, the direct calculation of the mass flow rate from chamber pressure and nozzle throat area is not possible for hybrid rockets because the characteristic exhaust velocity $c^*$ strongly depends on the oxidizer-to-fuel ratio. According to [21], the measurement of the instantaneous fuel mass flow rate for hybrid rockets is a great challenge, because it is a function of operation conditions, firing duration, port diameter, nozzle erosion conditions, etc. Therefore, the end-point averaging method based on the initial and final shapes of fuel charge is often used for estimating the average

fuel mass flow rate. However, the fuel mass flow rate is usually not constant during motor firings. The approach based on short firing duration for the end-point averaging is also not reliable due to the uncertainties introduced by ignition and shutdown transients [22]. Several reconstruction techniques for estimating the instantaneous fuel mass flow rate based on the measured chamber pressure and oxidizer mass flow rate have been reported in the literature on hybrid rockets [23–27]. However, all these techniques imply a known value of the oxidizer mass flow rate, whereas in the flow-through GGs, the latter can be unknown, especially when the GG is placed in the free approaching air stream. In the latter case, the combustion/gasification process in the GG can affect the mass flow rate of air at the GG intake.

This manuscript presents a method for the semi-empirical determination of the instantaneous rate of gas production in a flow-through GG placed in a free air stream with the allocation of a part of the gas flow produced by gasification of a low-melting solid material (LSM) in the total gas flow rate at the GG exit. This method and its demonstration are the novel and distinctive features of the present work.

## 2. Experimental Setup

Figure 1 shows the general view of a free jet facility for testing a GG in supersonic flow of gasifying agent (air). The facility was based on the Model Aerodynamic Facility (MAF) at the ITAM SB RAS [28]. Compressed air from high-pressure cylinders passed through a prechamber and entered a fire heater operating on combustion of hydrogen to obtain a hot air. Mass flow rate of hydrogen to the fire heater varied from 2 to 4 g/s. Apart from air and hydrogen, oxygen was added to the air flow in an amount of 20 to 40 g/s for keeping the mass fraction of oxygen at 23%wt in the vitiated air outflow. Thus, the outflowing gasifying agent contained oxygen (23%), nitrogen (75%), and steam (2%). The change in thermophysical properties of air due to steam admixture was neglected.

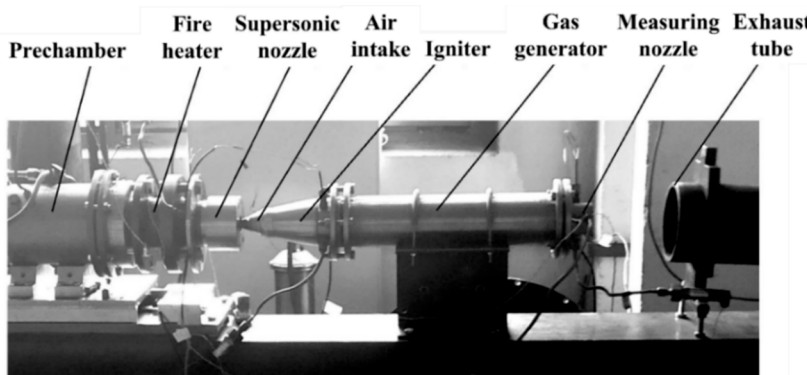

**Figure 1.** General view of the free jet test facility with GG.

Three profiled nozzles with an exit diameter of 30 mm designed for flow Mach numbers $M_0$ = 2.5, 3.0 and 4.0 could be attached to the fire heater. Table 1 shows the calculated and measured parameters of the corresponding jet flow. Measurements of the real values of the flow Mach number $M_1$ were performed using a Pitot pressure sensor (Pitot pressure $P_0{}'$) installed at the nozzle exit. In each experiment, the total pressure $P_0(t)$ and stagnation temperature $T_0(t)$ were recorded for calculating the mass flow rate of gases, $G_0(t)$. A jet of hot gasifying agent from the facility nozzle partly entered the GG mounted downstream along the facility axis.

**Table 1.** Characteristics of the jet flow at the entrance of GG intake.

| Designed $M_0$ | Mean $P_0'/P_0$ | Measured $M_1$ | $P_0$, MPa | $T_0$, K | $G_0$, kg/s |
|---|---|---|---|---|---|
| 2.5 | 0.528 | 2.43 | 1.90 | 665 | 0.85 |
| 3.0 | 0.346 | 2.94 | 3.20 | 830 | 0.79 |
| 4.0 | 0.163 | 3.81 | 3.00 | 850 | 0.32 |

## 3. Gas Generator Design

Figure 2 shows the schematic of the GG. An axisymmetric cylindrical intake with a diameter of 15 mm and length of 46.6 mm was installed in the frontal part of GG. From the cylindrical intake, hot gasifying agent flowed through a conical diffuser to enter the LSM sample. In the diffuser, a total pressure sensor and thermocouple were installed, which measured the stagnation pressure $P_{0,in}$ and stagnation temperature $T_{0,in}$ in the gas flow at the inlet to the LSM sample. Around the diffuser, a 36-g pyrotechnic igniter with a calorific value of about 1.5 MJ/kg was placed. It was used to ignite combustion of LSM aimed at providing heat for endothermic physical and chemical processes accompanying sample gasification. The igniter was triggered at a preset time by a command of the synchronization system. Typical igniter burning time was 0.10–0.15 s. Preliminary estimates showed that the gas temperature at the inlet of the LSM sample could theoretically reach 1300–2600 K, however the actual measured gas temperature was on the level of 1300–1500 K, thus indicating large heat loss into the walls.

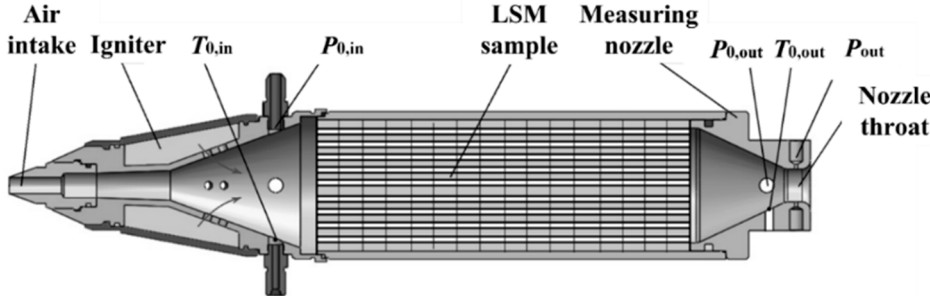

**Figure 2.** Schematic of flow-through GG.

The LSM sample was assembled from 16 identical polypropylene (PP) blocks 80 mm in diameter and 23 mm long (Figure 3). Each block had 61 orifices with a diameter of 4 mm. When assembling the sample, the orifices of all blocks were aligned to form straight longitudinal channels. The total initial sample mass was 1.28 kg. The masses of each block and sample in assembly were measured by balances both before and after test fires with an error of 0.05 g.

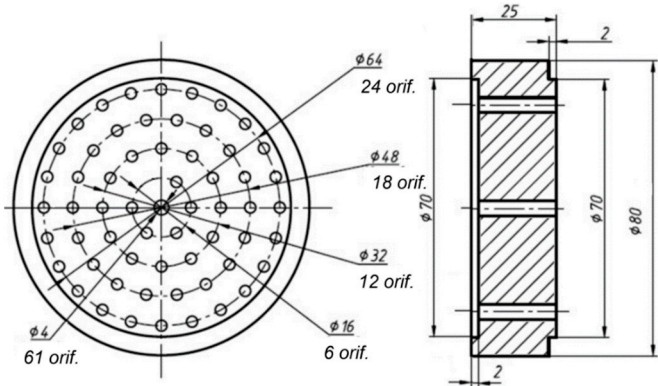

**Figure 3.** One of 16 identical polyethylene blocks for assembling a test sample; dimensions are given in millimeters.

A measuring nozzle with a throat of 13 or 20 mm in diameter was mounted at the GG outlet downstream of the LSM sample. In the nozzle, three thermocouples and total pressure sensor were installed to measure the stagnation pressure $P_{0,\text{out}}$ and stagnation temperature $T_{0,\text{out}}$. To measure static pressure $P_{\text{out}}$ and control the flow Mach number, a static pressure sensor was also mounted in the nozzle throat. After exiting the measuring nozzle, the exhaust jet was directed to the noise absorbing ventilation shaft at atmospheric pressure. For pressure measurements, pressure sensors RPD-I with a maximum pressure up to 1.0 and 2.5 MPa were used. The rated measurement error of the pressure sensors was 0.2%. The actual pressure measurement error was established by the results of numerous calibrations of the sensors in the expected pressure range. In this case, to control the set pressure, a PDE-020I reference pressure transducer was used with a basic relative error of $\pm 0.02\%$. The resulting estimate of the actual pressure measurement error did not exceed 1%. For temperature measurements, tungsten-rhenium thermocouples were used. To convert the electrical signal of thermocouples into temperature readings, a standard calibration table was used. The temperature measurement error did not exceed 5%.

The data acquisition system was based on the National Instruments NI PCI-6255 board. The system had 80 differential measuring channels. The bit depth of the analog-to-digital converter (ADC) was 16 bits. The maximum sampling frequency was 1.25 MHz. In the experiments, a sampling frequency of 1000 samples for each channel was used.

The frequency characteristics of the measuring equipment including pressure and temperature sensors as well as the data acquisition system were measured by applying preset pulse signals. It was shown based on the results of such dynamic tests that the operation frequency of the measuring equipment used ranged from 0 to 100 Hz. Since the characteristic frequencies of the processes under investigation did not exceed 10 Hz, dynamic measurement errors were not considered and were not taken into account.

## 4. Test Results

All test fires were conducted with igniter triggering. However, combustion of LSM sample failed in some test fires. Figures 4–6 show the typical pressure and temperature records in test fires without and with combustion of LSM samples. In test fires without combustion (tests 1, 4, and 7 in Figures 4–6), after ignition triggering, all measured parameters returned to their original values. In this case, the measured sample mass before and after test fires showed virtually no LSM depletion. In test fires with combustion (tests 2, 3, 5, 6, 8, 9 in Figures 4–6), the pressure at the LSM sample inlet and outlet after ignition triggering sharply increased by a factor of 1.5 to 2.0. The temperature at the sample outlet increased to 800–1000 K, although the temperature at the sample inlet corresponded to the temperature of the incoming gas flow and remained constant at 600–700 K. This indicated the existence of an exothermic oxidation process due to partial interaction of PP decomposition products with oxygen in the incoming flow of gasifying agent.

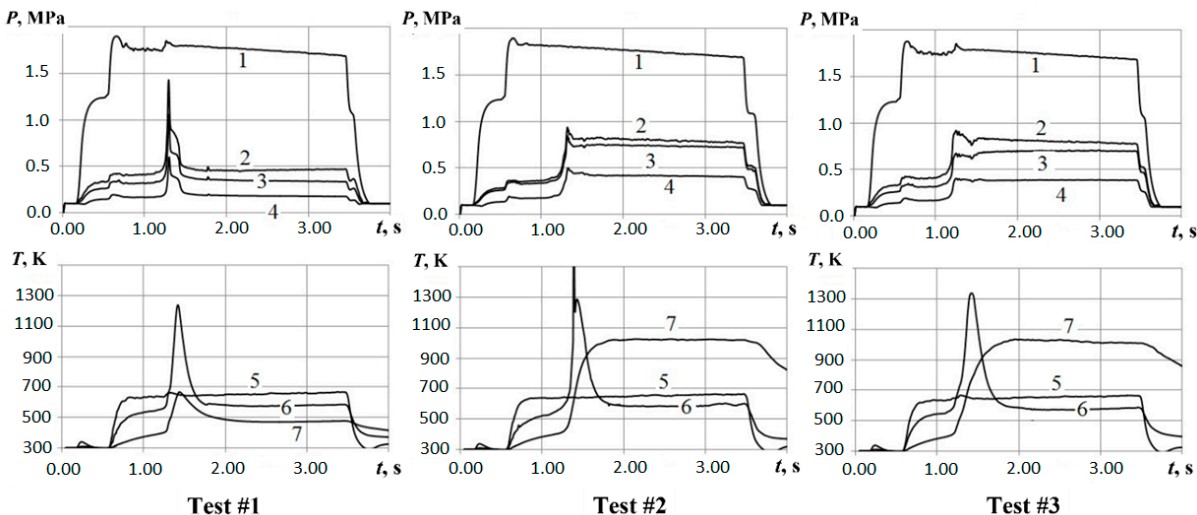

**Figure 4.** Test results at $M_1$ = 2.43 (see Table 1): 1—$P_0$; 2—$P_{0,in}$; 3—$P_{0,out}$; 4—$P_{out}$; 5—$T_0$; 6—$T_{0,in}$; 7—$T_{0,out}$.

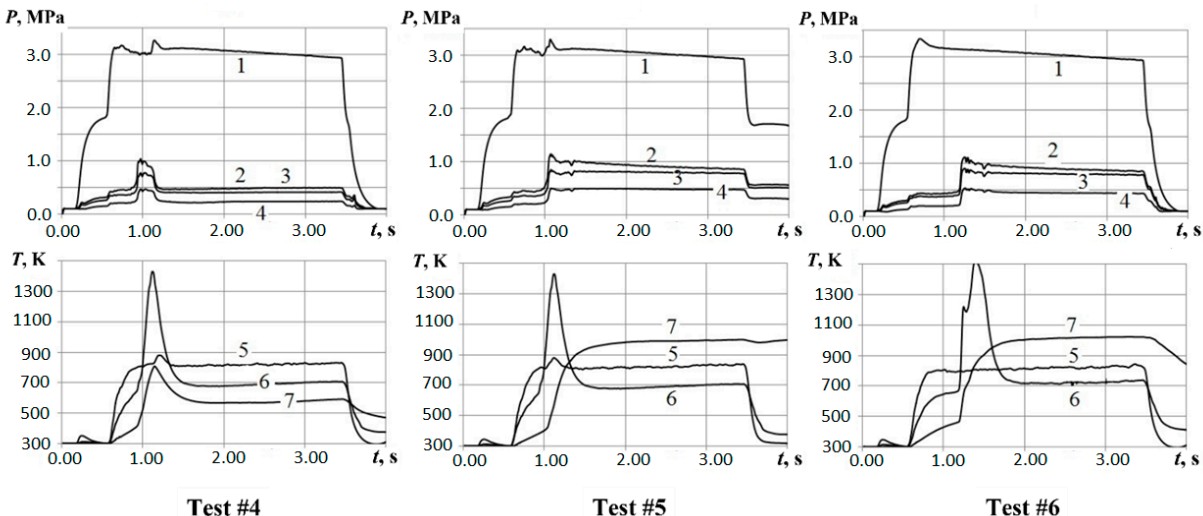

**Figure 5.** Test results at $M_1$ = 2.94 (see Table 1): 1—$P_0$; 2—$P_{0,in}$; 3—$P_{0,out}$; 4—$P_{out}$; 5—$T_0$; 6—$T_{0,in}$; 7—$T_{0,out}$.

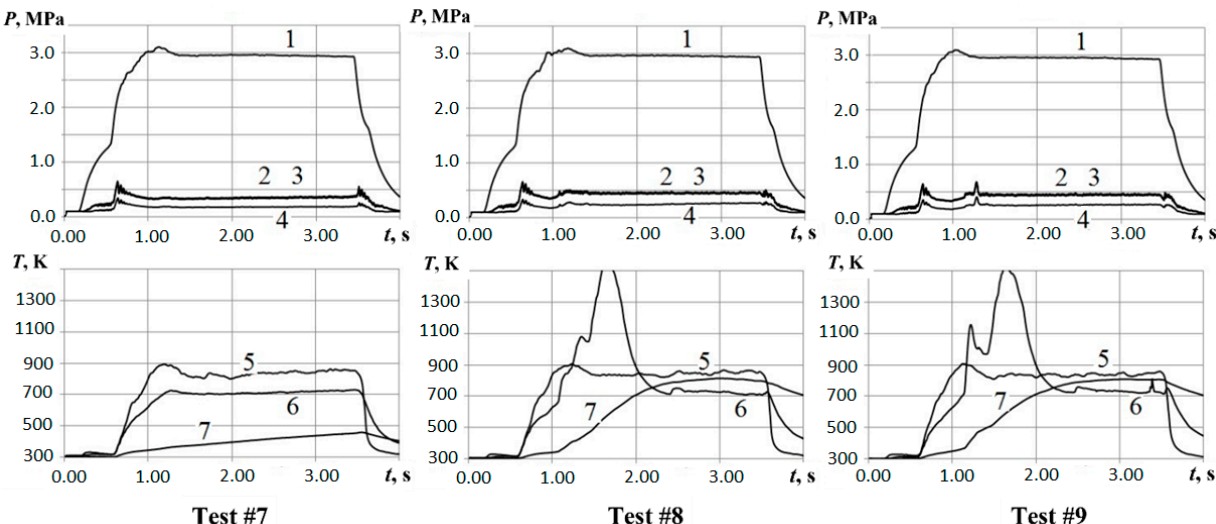

**Figure 6.** Test results at $M_1$ = 3.81 (see Table 1): 1—$P_0$; 2—$P_{0,\text{in}}$; 3—$P_{0,\text{out}}$; 4—$P_{\text{out}}$; 5—$T_0$; 6—$T_{0,\text{in}}$; 7—$T_{0,\text{out}}$.

## 5. Processing of Test Results

### 5.1. Calculation of Gas Flow Rate at GG Intake

The flow rate of gasifying agent at the GG intake was calculated based on the stagnation parameters $P_0(t)$ and $T_0(t)$ and the approaching flow Mach number $M_1$:

$$G_{\text{in}}(t) = \varphi F_{\text{in}}\rho(t)V(t) = \varphi F_{\text{in}}\, M_1 P_0(t)\pi(M_1)[\gamma R T_0(t)\tau(M_1)]^{1/2}[R T_0(t)\tau(M_1)]^{-1}$$
$$= \varphi F_{\text{in}}M_1\pi(M_1)\tau(M_1)^{-1/2}(\gamma/R)P_0(t)T_0(t)^{-1/2} \tag{1}$$

where $t$ is time, $\rho$ is gas density, $V$ is gas velocity, $F_{\text{in}}$ is the cross-sectional area of the GG intake, $\varphi$ is the contraction ratio of GG intake, $R$ = 287 kJ/kg/K is the gas constant for air, $\gamma$ = 1.4 is the specific heat ratio for air, $\pi(M_1)$ and $\tau(M_1)$ are the gas-dynamic functions.

### 5.2. Contraction Ratio of GG Intake

In test fires without combustion, a cylindrical GG intake cut out a stream of a cross-sectional area $F_{\text{in}}$ with the Mach number $M_1$ from the free jet emanating from the facility nozzle. In this case, the contraction ratio of the GG intake was $\varphi$ = 1.0. Let us jointly consider Figure 2 and curves 2 ($P_{0,\text{in}}$) in tests 1, 4, and 7 in Figures 4–6. Downstream of the cylindrical section of the intake, the supersonic gas flow entered the expanding conical diffuser and accelerated to Mach number $M_2$ ($M_2 > M_1$). Thereafter, due to geometric throttling in the LSM channels, the flow passed through a normal shock and became subsonic with a sharp decrease in the stagnation pressure to $P_{0,\text{in}}(M_2)$.

A similar flow pattern was observed in test fires with combustion prior to ignition triggering. Figure 7 shows several examples of such test fires in terms of the measured time histories of $P_{0,\text{in}}/P_0$ ratio. Prior to ignition triggering, the $P_{0,\text{in}}/P_0$ ratio corresponded to the total pressure loss at elevated Mach number $M_2 > M_1$. This indicated that the normal shock was located in the conical diffuser upstream of the total pressure sensor. After ignition triggering and establishment of LSM sample combustion, thermal throttling of the flow caused the normal shock to move upstream and exit the GG intake. Starting from this time instant, the $P_{0,\text{in}}/P_0$ ratio corresponded to the total pressure loss in the normal shock at the freestream Mach number $M_1$. A decrease in the Mach number ahead of the normal shock led, on the one hand, to a pressure rise in the flow entering the channels of LSM sample and thereby intensified the combustion and heat and mass transfer processes in sample channels. On the other hand, the head shock wave detached from the GG intake led to a decrease in the gas flow rate through the intake, i.e., to a decrease in the contraction ratio $\varphi$ in Equation (1). This posed a problem of determining the realistic value of the contraction ratio $\varphi$ < 1.0. The methodology for solving this problem by measuring the gas flow rate at

the GG outlet is presented below. Once the contraction ratio is determined, Equation (1) can be used for calculating the mass flow rate of gasifying agent entering the GG intake.

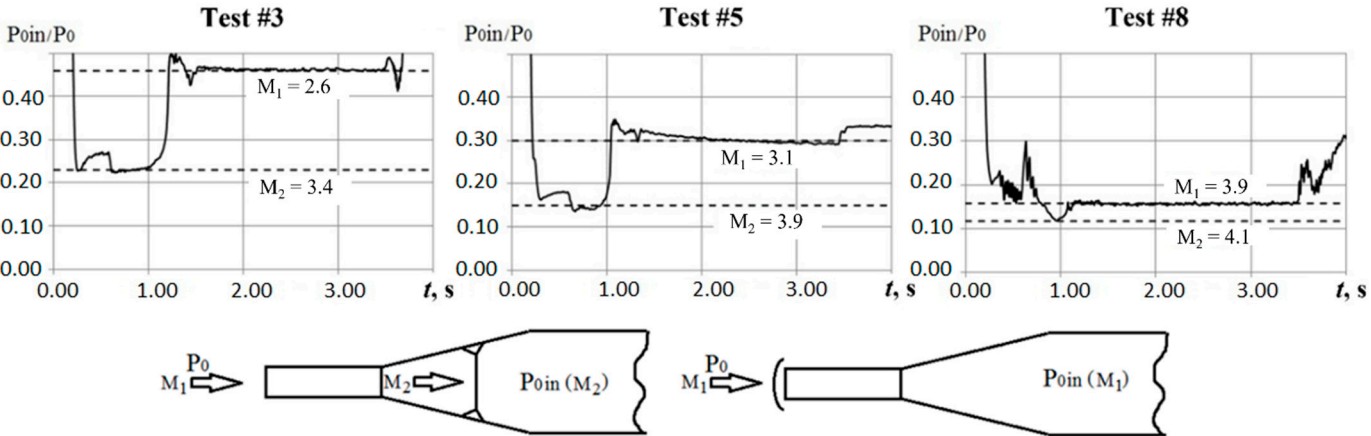

**Figure 7.** Formation of a diverted head shock at the GG intake in test fires with LSM sample combustion.

*5.3. Calculation of Gas Flow Rate at GG Outlet*

When the hot gasifying agent flows through the channels in the LSM sample, the LSM is heated, melted, pyrolyzed, and gasified. The produced gasification products are mixed and chemically react with the gasifying agent, thus leading to sample combustion accompanied with heat release. The heat of combustion promotes further sample gasification, and the gas flow rate increases. The task is to determine the gas flow rate at GG outlet.

In the test fires, this gas flow rate was determined by mounting a sonic nozzle of cross-sectional area $F^*$ (asterisk * denotes sonic nozzle throat) at the GG outlet and measuring the values of stagnation pressure $P_{0,out}$ and stagnation temperature $T_{0,out}$ before the nozzle. The values of $F^*$, $P_{0,out}$, and $T_{0,out}$ were used to calculate $G_{out}$ using the relation:

$$G_{out}(t) = mF^*P_{0,out}(t)T_{0,out}(t)^{-1/2} \tag{2}$$

where the dimensional coefficient $m$ is a function of gas composition at the nozzle:

$$m = [(\gamma^*/R^*)(2/(\gamma^*+1))^{(\gamma^*+1)/(\gamma-1)}]^{1/2} \tag{3}$$

The main difficulty in calculating $G_{out}$ is then the determination of coefficient $m$, as the gas composition depends on the LSM combustion completeness, as well as local instantaneous pressure and temperature. Herein, the values of $\gamma^*$ and $R^*$ for the gas mixture were determined by iterations using the Astra 4 thermodynamic code [29].

As an example, Figure 8 shows the results of thermodynamic calculations for mixtures composed of PP and air with the PP-to-air mass ratio ranging from 0 to 50% at a constant pressure of 1 MPa. Note that the results of calculations in a pressure range between 0.1 and 1 MPa differ by less than 3% only.

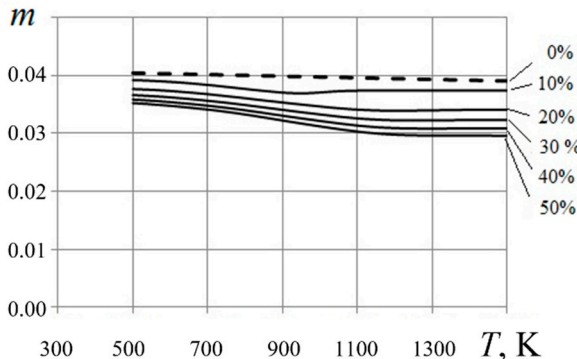

**Figure 8.** Calculated dependences of coefficient *m* on temperature for PP–air mixtures with different content of PP (mass basis) at pressure 1 MPa.

### 5.4. Flow Rate of Polypropylene Gasification Products

Figures 9–11 show the time histories of flow rates $G_{in}$ and $G_{out}$ calculated based on Equations (1) and (2), respectively, for different freestream Mach numbers $M_1$. In addition to plots $G_{in}(t)$ and $G_{out}(t)$, the calculated difference between these flow rates, $\Delta G = G_{out}(t) - G_{in}(t)$ is also plotted in Figure 9 to Figure 11. Clearly, this difference corresponds to the total flow rate of product gases generated by LSM sample gasification.

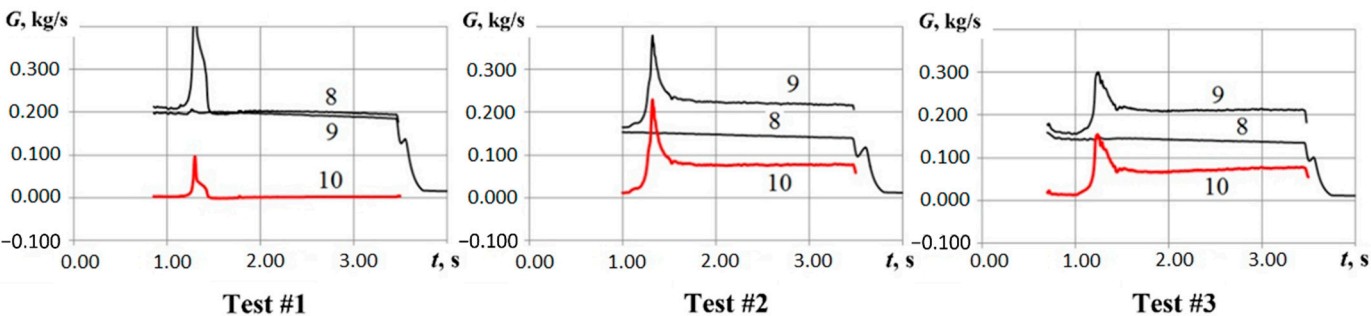

**Figure 9.** Calculated time histories of flow rates $G_{in}$ (curve 8), $G_{out}$ (9) and $\Delta G = G_{out}(t) - G_{in}(t)$ (10) at $M_1 = 2.43$ in tests 1 to 3.

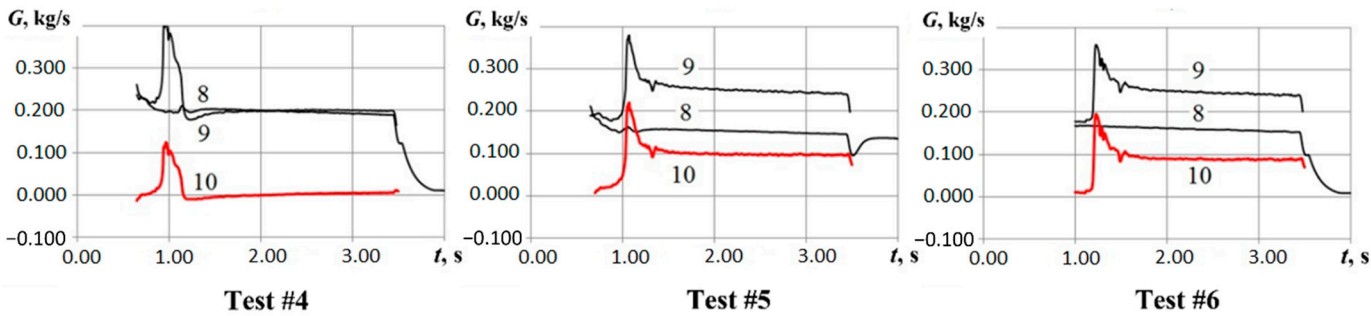

**Figure 10.** Calculated time histories of flow rates $G_{in}$ (curve 8), $G_{out}$ (9) and $\Delta G = G_{out}(t) - G_{in}(t)$ (10) at $M_1 = 2.94$ in tests 4 to 6.

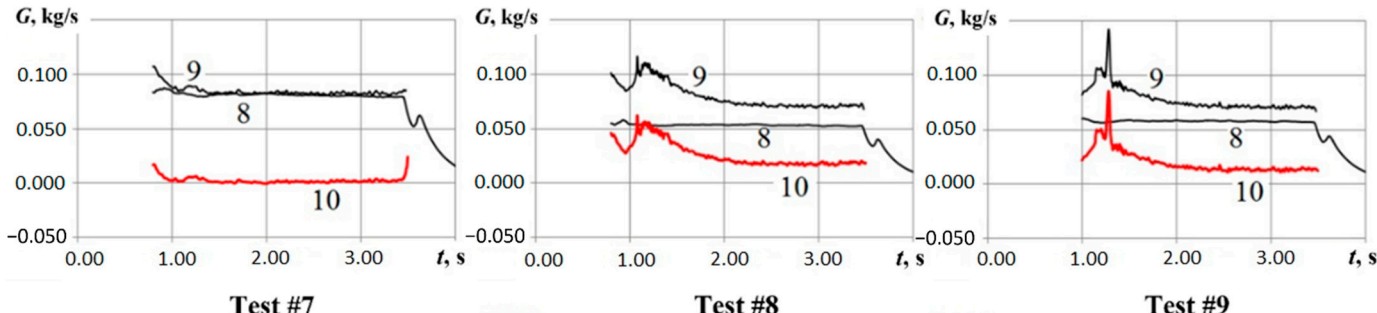

**Figure 11.** Calculated time histories of flow rates $G_{in}$ (curve 8), $G_{out}$ (9) and $\Delta G = G_{out}(t) - G_{in}(t)$ (10) at $M_1 = 3.81$ in tests 7 to 9.

As mentioned above, LSM combustion in tests 1, 4, and 7 was absent. Therefore, the flow rates $G_{in}$ and $G_{out}$ were virtually equal. In test fires with combustion, a noticeable LSM gasification accompanied with sample combustion was detected. In these latter test fires, the gas flow rates $G_{in}$ and $G_{out}$ differed markedly. According to Figures 9–11 the flow rate of gasification products ($\Delta G = G_{out}(t) - G_{in}(t)$) in test fires with combustion could be constant, like in Figures 9 and 10, or variable in time, like in Figure 11. The mean values of the flow rates of gasification products in Figures 9–11 took the values of 0.08 kg/s at $M_1 = 2.43$, 0.10 kg/s at $M_1 = 2.94$, and 0.05–0.02 kg/s at $M_1 = 3.81$.

An important element of test data processing procedure was the determination of LSM sample mass before ($W_1$) and after ($W_2$) test fire. The difference between these masses ($\Delta W = W_1 - W_2$) determines the amount of gasified sample material leaving the GG with exhaust gases. Based on the calculated flow rates $G_{in}$ and $G_{out}$ and measured sample masses $W_1$ and $W_2$, one can derive the following balance equation for determining the intake contraction ratio $\varphi$:

$$W_1 - W_2 = mF^* {}_{t1}\!\int^{t2} P_{0,out}(\tau) T_{0,out}(\tau)^{-1/2} d\tau - \varphi F_{in} M_1 \pi(M_1)\tau(M_1)^{-1/2}(\gamma/R) {}_{t1}\!\int^{t2} P_0(\tau)T_0(\tau)^{-1/2} d\tau \quad (4)$$

In Equation (4), time $t_1$ corresponds to the start of gas temperature rise after igniter triggering and time $t_2$ corresponds to the shutdown of air supply. An unknown value of the air intake contraction ratio $\varphi$ can be now determined from Equation (4). Once $\varphi$ is determined, the realistic value of gasifying agent flow rate at the inlet to LSM sample can be obtained. Furthermore, the value of coefficient $m$ in Equation (3) can be refined.

Table 2 shows the test results thus obtained and refined. Now, the time integrals of functions $\Delta G(t)$ shown in Figures 9–11 are equal to the corresponding values of $\Delta W$ in each test fire. As seen from Table 2, the ratio of total amounts of gasifying agent to gasification products, $\int G_{in}dt/\int\Delta Gdt$, varied from 1.61 to 2.86 in the test fires. This means that for obtaining 1 kg of gasification products one consumes 1.61 to 2.86 kg of gasifying agent.

**Table 2.** Results of tests with combustion.

| Test | $\varphi$ | $\int G_{in}dt$, kg | $m$ | $\Delta W$, kg | $\int\Delta Gdt$, kg | $\int G_{in}dt/\int\Delta Gdt$ |
|------|-----------|---------------------|-----|----------------|----------------------|-------------------------------|
| 2 | 0.751 | 0.330 | 0.0306 | 0.187 | 0.186 | 1.77 |
| 3 | 0.728 | 0.326 | 0.0308 | 0.176 | 0.174 | 1.86 |
| 5 | 0.774 | 0.444 | 0.0311 | 0.276 | 0.273 | 1.61 |
| 6 | 0.810 | 0.372 | 0.0312 | 0.219 | 0.216 | 1.70 |
| 8 | 0.662 | 0.133 | 0.0344 | 0.064 | 0.064 | 2.10 |
| 9 | 0.717 | 0.144 | 0.0344 | 0.050 | 0.050 | 2.86 |

## 6. Amendment

The adopted technology of dividing the LSM sample into separate blocks and measuring their individual masses $W_i$ before and after test fires allowed obtaining additional useful information on the zones of maximum LSM decomposition along the sample length. Figure 12 shows the distribution of masses $W_i$ of individual blocks in a test sample, measured before and after test 3. The individual blocks are numbered from the inlet of the LSM sample. As seen, the zone of most intense LSM decomposition in this test was located between blocks 4 and 9. Figure 13 shows the photographs of all individual blocks of the LSM sample before and after test 3.

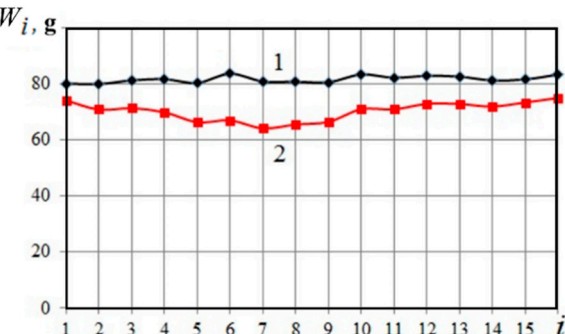

**Figure 12.** Masses of blocks composing LSM sample before (1) and after (2) test 3 with combustion. Burning time $t_2 - t_1 = 3.46 - 1.15 = 2.31$ s.

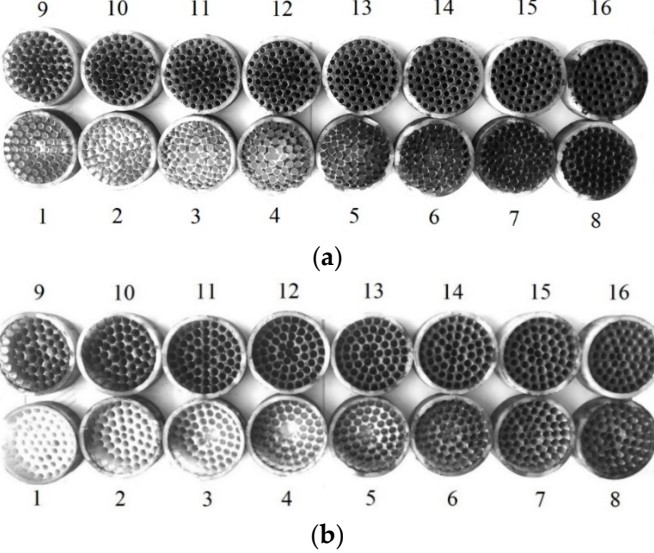

**Figure 13.** Photographs of (**a**) upstream and (**b**) downstream faces of blocks composing LSM sample after test 3 with combustion.

## 7. Conclusions

A novel semi-empirical method is proposed for evaluating the instantaneous mass flow rate of gasification products in the flow-through gas generators operating on air-assisted gasification of low-melting solid materials. The instantaneous mass flow rate of gasification products is derived based on determining the mass flow rate of air captured by the gas generator intake in a supersonic flow and the instantaneous mass flow rate of gases at the gas generator outlet, supplemented with measurements of sample mass before and after test firing.

To verify the methodology, a set of test fires with hot air as gasifying agent and polypropylene as a low-melting solid material were conducted under conditions of incident

supersonic flow of gasifying agent with Mach numbers 2.43, 2.94, and 3.81 and stagnation temperatures 600–700 K. The mean flow rates of gasification products obtained experimentally were 0.08 kg/s at Mach 2.43, 0.10 kg/s at Mach 2.94, and 0.05–0.02 kg/s at Mach 3.81. In the test fires, the ratio of total amounts of air to gasification products varied from 1.61 to 2.86 (mass basis). This means that for obtaining 1 kg of gasification products one needs 1.61 to 2.86 kg of air.

**Author Contributions:** V.I.Z.: Conceptualization, methodology; D.A.V.: investigation; D.G.N.: investigation; S.M.F.: writing original draft and editing; project administration. All authors have read and agreed to the published version of the manuscript.

**Funding:** This work was financially supported by the Ministry of Science and Higher Education of Russian Federation under state contract N13.1902.21.0014 (agreement N075-15-2020-806).

**Institutional Review Board Statement:** Not applicable.

**Informed Consent Statement:** Not applicable.

**Data Availability Statement:** The data presented in this study are available on request from the corresponding author.

**Conflicts of Interest:** The authors declare no conflict of interest.

## Abbreviations and Nomenclature

| | |
|---|---|
| ADC | Analog-to-digital converter |
| GG | Gas generator |
| LSM | Low-melting solid material |
| MAF | Model Aerodynamic Facility |
| ITAM SB RAS | Khristianovich Institute of Theoretical and Applied Mechanics of the Siberian Branch of the Russian Academy of Sciences |

| | | |
|---|---|---|
| $c^*$ | Characteristic exhaust velocity | m/s |
| $M_0$ | Design nozzle Mach number | - |
| $M_1$ | Air flow Mach number at the nozzle exit | - |
| $M_2$ | Mach number before the normal shock in diffuser | - |
| $P_0'$ | Pitot pressure at the nozzle exit | Pa |
| $P_0$ | Total pressure of the air flow | Pa |
| $T_0$ | Total (stagnation) temperature of the flow | K |
| $G_0$ | Mass flow rate | kg/s |
| $P_{0,in}$ | Total pressure at the entrance of LSM sample | Pa |
| $T_{0,in}$ | Total temperature at the entrance of LSM sample | K |
| $P_{0,out}$ | Total pressure at the exit of LSM sample | Pa |
| $T_{0,out}$ | Total temperature at the exit of LSM sample | K |
| $P_{out}$ | Static pressure at the sound nozzle | Pa |
| $F_{in}$ | Area of intake entrance cross-section | $m^2$ |
| $F^*$ | Area of sonic nozzle throat cross-section | $m^2$ |
| $G_{in}$ | Air flow rate through intake | kg/s |
| $G_{out}$ | Gas flow rate at the sonic nozzle throat | kg/s |
| $\varphi$ | Contraction ratio of intake | - |
| $R$ | Gas constant for air | J/kg/K |
| $R^*$ | Gas constant for gas mixture | J/kg/K |
| $\gamma$ | Specific heat ratio for air | - |
| $\gamma^*$ | Specific heat ratio for gas mixture | - |
| $\pi(M1)$ | Gas-dynamic function | - |
| $\tau(M1)$ | Gas-dynamic function | - |
| $m$ | Dimensional coefficient | - |
| $W_1$ | Sample mass before test | kg |
| $W_2$ | Sample mass after test | kg |
| $W_i$ | Masses of individual blocks in the LSM sample | g |

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
