# Peer review of "Measurement of Gas Flow Rate at Gasification of Low-Melting Materials in a Flow-Through Gas Generator"

_energies, doi:10.3390/en15155741_

Round 1

Reviewer 1 Report

This work presents an experimental study on the measurement of gas flow with high Mach numbers. It is interesting. However, there are still some problems needed to be improved.

1) The test results are transient and dynamic. Further discussion about the transient characteristics should be given.

2) What is the matter with page 7? It should be arranged.

3) The conclusions are not clear. It should be rewritten.

Author Response

We are grateful to the reviewer for valuable comments. We made our best to follow all the comments. All changes in the revised manuscript are marked in green.

This work presents an experimental study on the measurement of gas flow with high Mach numbers. It is interesting. However, there are still some problems needed to be improved.

1) The test results are transient and dynamic. Further discussion about the transient characteristics should be given.

To address this comment, we have added the following paragraph to the text:

“The frequency characteristics of the measuring equipment including pressure and temperature sensors as well as the data acquisition system were measured by applying preset pulse signals. It was shown based on the results of such dynamic tests, that the operation frequency of the measuring equipment used ranged from 0 to 100 Hz. Since the characteristic frequencies of the processes under investigation did not exceed 10 Hz, dynamic measurement errors were not considered and were not taken into account.”

2) What is the matter with page 7? It should be arranged.

Sorry, something was wrong with the page header in the manuscript template.

3) The conclusions are not clear. It should be rewritten.

To address this comment, we have reformulated and shortened the Conclusions and replaced “gasification agent” by “air.”

Reviewer 2 Report

The authors used a semi-empirical method to determine the rate of gas production flow, through gas generator. The paper is generally well prepared and can be accepted after minor revisions:

 The introduction is very short and needs to be extended, by presenting and discussing earlier published related to the subject.

The authors must clearly state the novelty of their work.

Higher resolutions are to be provided for the figures.

An experimental uncertainty study is to be performed.

The data acquisition system is to be described.

The titles of figs 4 to 6 are to be revised, it to be clearly indicated which fig is corresponding to which parameters.

A nomenclature is to be added.

Author Response

We are grateful to the reviewer for valuable comments. We made our best to follow all the comments. All changes in the revised manuscript are marked in yellow.

The authors used a semi-empirical method to determine the rate of gas production flow, through gas generator. The paper is generally well prepared and can be accepted after minor revisions:

The introduction is very short and needs to be extended, by presenting and discussing earlier published related to the subject.

To address this comment, we have added the following text to the Introduction section and added 7 more references to the list of references:

“A good illustrative example is the GG of a hybrid rocket. Contrary to solid rockets, the direct calculation of the mass flow rate from chamber pressure and nozzle throat area is not possible for hybrid rockets because the characteristic exhaust velocity c* strongly depends on the oxidizer-to-fuel ratio.  According to [21], the measurement of the instantaneous fuel mass flow rate for hybrid rockets is a great challenge, because it is a function of operation conditions, firing duration, port diameter, nozzle erosion conditions, etc. Therefore, the end-point averaging method based on the initial and final shapes of fuel charge is often used for estimating the average fuel mass flow rate. However, the fuel mass flow rate is usually not constant during motor firings. The approach based on short firing duration for the end-point averaging is also not reliable due to the uncertainties introduced by ignition and shutdown transients [22]. Several reconstruction techniques for estimating the instantaneous fuel mass flow rate based on the measured chamber pressure and oxidizer mass flow rate were reported in the literature on hybrid rockets [23–27]. However, all these techniques imply a known value of the oxidizer mass flow rate, whereas it the flow-through GGs, the latter can be unknown, especially when the GG is placed in the free approaching air stream. In the latter case, the combustion/gasification process in the GG can affect the mass flow rate of air at the GG intake.”

1). The authors must clearly state the novelty of their work.

To address this comment, we have reformulated the last paragraph in Introduction:

“This manuscript presents a method for the semi-empirical determination of the instantaneous rate of gas production in a flow-through GG placed in a free air stream with the allocation of a part of the gas flow produced by gasification of a low-melting solid material (LSM) in the total gas flow rate at the GG exit. This method and its demonstration are the novel and distinctive features of the present work.”

2). Higher resolutions are to be provided for the figures.

To address this comment, we have increased the font sizes in all figures.

3). An experimental uncertainty study is to be performed.

To address this comment, we have added the following paragraph to the text:

“The factory measurement error of the pressure sensors was 0.2%. The actual pressure measurement error was established by the results of numerous calibrations of the sensors in the expected pressure range. In this case, to control the set pressure, a PDE-020I reference pressure transducer was used with a basic relative error of ±0.02%. The resulting estimate of the actual pressure measurement error did not exceed 1%. For temperature measurements, tungsten-rhenium thermocouples were used. To convert the electrical signal of thermocouples into temperature readings, a standard calibration table was used. The temperature measurement error did not exceed 5%.”

4).The data acquisition system is to be described.

To address this comment, we have added the following paragraph to the text:

“The data acquisition system was based on the National Instruments NI PCI-6255 board. The system had 80 differential measuring channels. The bit depth of the analog-to-digital converter (ADC) was 16 bits. The maximum sampling frequency was 1.25 MHz. In the experiments, a sampling frequency of 1000 samples for each channel was used.”

5). The titles of figs 4 to 6 are to be revised, it to be clearly indicated which fig is corresponding to which parameters.

To address this comment, we have added the reference to Table 1 in the captions of figs 4 to 6.

6).A nomenclature is to be added.

To address this comment, we have added the lists of Abbreviations and Nomenclature at the end of the manuscript.

Round 2

Reviewer 1 Report

I think it is ok now.